# Bond Wire Damage Detection Method on Discrete MOSFETs Based on Two-Port Network Measurement

**DOI:** 10.3390/mi13071075

**Published:** 2022-07-07

**Authors:** Minghui Yun, Miao Cai, Daoguo Yang, Yiren Yang, Jing Xiao, Guoqi Zhang

**Affiliations:** 1School of Mechanical and Electrical Engineering, Guilin University of Electronic Technology, Guilin 541004, China; yunminghui_01@163.com (M.Y.); yangyirentrain@126.com (Y.Y.); xiaojing@guet.edu.cn (J.X.); g.q.zhang@tudelft.nl (G.Z.); 2Department of Microelectronics, Delft University of Technology, Mekelweg 4, 2628 CD Delft, The Netherlands

**Keywords:** MOSFET, bond wire fault, two-port network, source parasitic inductance

## Abstract

Bond wire damage is one of the most common failure modes of metal-oxide semiconductor field-effect transistor (MOSFET) power devices in wire-welded packaging. This paper proposes a novel bond wire damage detection approach based on two-port network measurement by identifying the MOSFET source parasitic inductance (*L*_S_). Numerical calculation shows that the number of bond wire liftoffs will change the *L*_S_, which can be used as an effective bond wire damage precursor. Considering a power MOSFET as a two-port network, *L*_S_ is accurately extracted from frequency domain impedance (*Z*−parameter) using a vector network analyzer under zero biasing conditions. Bond wire cutoff experiments are employed to validate the proposed approach for bond wire damage detection. The result shows that *L*_S_ increases with the rising severity of bond wire faults, and even the slight fault shows a high sensitivity, which can be effectively used to quantify the number of bond wire liftoffs of discrete MOSFETs. Meanwhile, the source parasitic resistance (*R*_S_) extracted from the proposed two-port network measurement can be used for the bond wire damage detection of high switching frequency silicon carbide MOSFETs. This approach offers an effective quality screening technology for discrete MOSFETs without power on treatment.

## 1. Introduction

Power electronic devices are widely used in mission-critical applications, such as locomotive traction, high-speed railway, electric vehicles, industrial frequency conversion, and renewable energy generation [1,2]. Literature studies indicated that the failure rate of power electronic devices among all converter failure types is 31%, accounting for the largest proportion among all failure types [3,4]. High-power metal-oxide semiconductor field-effect transistor (MOSFET) is one of the most critical and fragile elements in power electronic devices operating in harsh and uncertain conditions. MOSFETs will suffer from continuous excessive electrical–thermal–mechanical stresses and damage the bond wires and the solder layer. The reliability of MOSFETs has been attracting increasing research interest. In practical applications, the fatigue failure of power semiconductor devices in wire welding packaging is mostly manifested as the bond wires completely liftoff [1,5,6,7,8]. Therefore, the quality detection of MOSFET bond wire is of considerable importance to avoid the catastrophic failure of power electronic converters in the lifecycle.

Middle- and high-power MOSFETs are generally based on wire-welded packaging, which uses some parallel aluminum bond wires to improve the current carrying capacity for electrical interconnections between drain and source terminals. This condition introduces a problem that cannot be ignored; that is, the slight fault of bond wires will not immediately affect the performance of MOSFETs, which increases the difficulty of bond wire damage detection. Numerous research papers indicated that the commonly used bond wire reliability detection methods can be classified into two: degradation precursor- and morphology characteristic-based methods. Degradation precursor-based methods usually predict bond wire damage by measuring different types of signals and comparing them with the healthy device. These methods can be classified into the following three classes according to the type of signals used: voltage, current, and other signal precursor-based approaches. The first class is the voltage precursor-based approach. On-state drain-source (*V*_DS_) [8], collector–emitter saturated voltage (*V*_CE(sat)_) [9], gate threshold voltage (*V*_th_) [10], and turn-on gate voltage overshoot [5] are selected as bond wire fault indicators. Measuring the voltage signal is usually easy, and the sensitivity is minimal when the bond wire faults are minor. Moreover, the accuracy of the measurement is easily affected by the changes in bus voltage, current, and chip junction temperature; thus, strictly ensuring the high consistency of test conditions in each measurement is necessary. The second class is the current precursor-based approach. Gate current (*I*_G_) [11] and short-circuit current (*I*_SC_) [12,13] are usually selected as bond wire fault indicators. Reference [14] indicates that *I*_SC_ is sensitive to bond wire failures but requires accurate control of gate-drive voltage and junction temperature (*T*_j_) in measurement, which adds complexity to applications. Reference [15] found that the bond wire liftoff has minimal influence on the dynamic differences of *I*_G_. A high-precision A/D converter with a high sampling rate is needed to improve sampling precision, however using such a converter in the gate driver is too expensive. The third class is the other signal precursor-based approaches, including junction temperature (*T*_j_) [16], thermal resistance from junction to case (*R*_th_) [17], on-resistance (*R*_DS(on)_) [18], Miller plateau duration (*V*_GP_) [19], and others. *R*_DS(on)_ and *R*_th_ belong to temperature-dependent parameters, and the measurement accuracy is affected by the junction temperature, *T*_j_. Unfortunately, *T*_j_ cannot be measured directly, and the stable control of *T*_j_ is also a technical problem. Miller plateau duration has minimal sensitivity when one or two bond wires liftoff. Morphology characteristic-based methods mainly include thermal imaging [20,21,22,23] (eddy current pulse thermal imaging, infrared imaging) and structural imaging [24,25,26] (X-ray imaging, ultra-sound imaging, and industrial computerized tomography). Thermal imaging can identify the location of potential damage by observing the surface temperature distribution of the power devices, which is mainly used to detect solder-layer defects. Structural imaging is a non-destructive testing technology and can directly detect inner defects of devices by identifying the phase and amplitude of the reflected signals. However, the key to obtaining the ideal imaging quality lies in the accurate height estimation of the bond wire in the Z-axis direction in advance by the users, which is a remarkably difficult and time-consuming task. If delamination exists between the epoxy molding compound (EMC) layer and the upper surface of the die, then penetrating the EMC for effective bond wire imaging is difficult for the pulse ray, resulting in a limited application.

A novel bond wire damage detection approach based on two-port network measurement by identifying the MOSFET source parasitic inductance (*L*_S_) is proposed in this study. Based on frequency domain impedance analysis, MOSFET is equivalent to some second-order *RLC* circuits comprising independent inductances, capacitances, and resistances in series, whereas the high-frequency impedance of MOSFET is dominated by the inductive components. Therefore, the bond wire is equivalent to the pure inductance model at a frequency much higher than the self-resonant frequency (*f*_SRF_). The liftoff of the bond wire will increase the total parasitic inductance of the parallel bond wires, resulting in increasing high-frequency impedance. This notion provides a new idea that the physical failure of the bond wire can be mapped to the change in high-frequency impedance. Considering a power MOSFET as a two-port network, *L*_S_ is accurately extracted from vector network analyzer (VNA) measurement under zero basing conditions. The positive correlation between the *L*_S_ and the number of bond wire faults is then determined. The experimental results reveal that even a slight bond wire fault can be detected with high resolution. This method offers an effective bond wire damage detection technology for power-discrete devices without power on treatment.

## 2. Methodologies

### 2.1. MOSFET Small-Signal Equivalent Circuit

The schematic of a cross-section of a half-vertical-diffused MOSFET with a package structure is illustrated in Figure 1a. An ideal MOSFET chip can be equivalent to the constant and variable active devices, which comprise the voltage-controlled current source, internal parasitic capacitances, and anti-parallel body diode. The internal parasitic capacitances include drain–source capacitance (*C*_DS_), gate–source capacitance (*C*_GS_), and gate–drain capacitance (*C*_GD_). The chip and external terminals are electrically interconnected through the aluminum bond wire packaging technology. Additional parasitic parameters are inevitably introduced: (1) *L*_S-ter_ and *L*_G-ter_, *L*_D-ter_ and *R*_S-ter_, and *R*_G-ter_ and *R*_D-ter_ are generated from the gate, source, and drain terminals, respectively. (2) *L*_S-BW_, *L*_G-BW_, and *R*_S-BW_, *R_G_*_-BW_ are generated from bond wires. (3) *R*_D-solder_ and *R*_D-base_ are generated from the solder and baseplate layers, respectively. The small-signal equivalent circuit model is simplified. The parasitic inductances are combined into *L*_G_, *L*_S_, and *L*_D_ and the parasitic resistors into *R*_S_, *R*_G_, and *R*_D_ to facilitate the analysis, as shown in Figure 1b. Figure 2 depicts a typical power MOSFET in a TO−247 discrete package, which is encapsulated by a chip, copper substrate, bond wires, lead terminals, solder layer, and an EMC. The equivalent circuit of the MOSFET power device is decomposed in sequence according to the drain-to-source current loop.

### 2.2. Bond Wire Parasitic Inductance

MOSFETs in wire-welding packaging comprise multilayered materials with different coefficients of thermal expansion (CTE). Long-term thermal stress causes the expansion of different materials at various rates, specifically in the weak points of the wire bond root, resulting in bond wire fatigue and degradation. Figure 3 shows a 3D structural diagram of the four parallel bond wires, wherein each bond wire can be equivalent to a set of *RL* series circuits. The partial inductance, *L*_Bond_, of a bond wire comprises partial self and mutual inductances. According to [27], the self-inductance, *L* (in nH), of a single bond wire can be extracted by the simplified Equation (1), and the parasitic mutual-inductance, *M* (in nH), of a single bond wire can be determined by the simplified Equation (2).
(1)L=5l× [(In(4l/d)−0.75)],
(2)$$M=5\;\times \;[\mathrm{In}(\frac{2l}{s})\;-\;1\;+\;\frac{s}{l}\;-\;{\left(\frac{s}{2l}\right)}^{2}],$$
where *l* stands for the length of the bond wire (in inches), *d* stands for the diameter of the bond wire (in inches), and *s* stands for the distance between two bond wire centers (in inches). The mutual inductance for multiple bond wires is caused by the magnetic coupling of multiple bond wires. The internal structure and bond wire dimension of IXFK32N100P (IXYS Corporation, Milpitas, CA, USA and Leiden, The Netherlands) and C2M0160120D (Wolfspeed Corporation, Durham, NC, USA) are shown in Figure 4 and Table 1. Each bond wire is marked with a serial number for easy identification.

For IXFK32N100P, the parasitic inductance of bond wire No. 1, *L*_Bond_1_, lies in the addition of bond wire self-inductance *L*_1_ and mutual inductances *M*_12_, *M*_13_, *M*_14_, *M*_15_, and *M*_16_. Similarly, the parasitic inductance of bond wire No. 2, *L*_Bond_2_, lies in the addition of bond wire self-inductance *L*_2_ and mutual inductances *M*_21_, *M*_23_, *M*_24_, *M*_25_, and *M*_26_. The parasitic inductance of other bond wires is calculated by the same method. Therefore, the bond wire parasitic inductance, *L*_Bond_*m*_, of No. *m* is the sum of bond wire self-inductance *L_m_* of No. *m* (calculated by Equation (1)) and bond wire mutual inductances *M_mn_* between No. *m* and No. *n* (calculated by Equation (2)), which can be described as follows:(3)$$L_{\mathrm{Bond}\_m\;}{=L}_{m\;}+\overset{n}{\underset{n=1}{\sum }}M_{mn}\;(m\;\neq \;n)$$

Figure 4 shows that the multiple bond wires of discrete MOSFETs are not strictly parallel to each other. To simplify the calculation, the non-parallel distribution of the parallel connection bond wires between the source and source terminal is ignored, and the average spacing between the two bond wires and the length of the short bond wire is selected for mutual inductance estimation. The parasitic inductance, *L*_Bond_, is calculated and shown in Table 2. The total parasitic inductances of IXFK32N100P, *L*_Bond_IX_, and C2M0160120D, *L*_Bond_C2_, are correspondingly 4.33 and 3.75 nH, which are the results of parallel connections of six and four inductances, respectively.

### 2.3. Two-Port Parasitic Inductance Extraction Approach

The two-port scattering (*S*) parameter measurement with VNA is used in this study to extract the parasitic parameters of MOSFET [28,29]. The MOSFET small-signal equivalent circuit under zero biasing conditions, which is a two-port network with S–G and D–G as Ports 1 and 2, respectively, is shown in Figure 5. Each of the two-port network Z−parameters, *Z*_11_, *Z*_12_, *Z*_21_, and *Z*_22_, can be equivalent to some second-order *RLC* circuits comprising independent inductances, capacitances, and resistances in series. *Z*_11_ is equivalent to the *L*_S_–*R*_S_–*C*_S_–*C*_G_–*R*_G_–*L*_G_ series circuit, *Z*_12_ and *Z*_21_ are equivalent to the same *G*_G_–*R*_G_–*L*_G_ series circuit, and *Z*_22_ is equivalent to the *L*_G_–*R*_G_–*C*_G_–*C*_D_–*R*_D_–*L*_D_ series circuit. Notably, as a standard form of the two-port network, the equivalent capacitances (*C*_G_, *C*_D_, and *C*_S_) demonstrate a star connection in Figure 5, which is different from the delta connection of parasitic capacitances (*C*_GS_, *C*_DS_, and *C*_GD_) in Figure 1b. Therefore, the star connection should be converted into a delta connection by using Equations (4)–(6) to extract the parasitic capacitances.
(4)$$C_{\mathrm{GS}}{\;=\;C}_{G}C_{S}\;/{(C}_{G}{+C}_{D}\;{+C}_{S}),$$
(5)$$C_{\mathrm{GD}}\;=\;C_{G}C_{D}\;/{(C}_{G}{+C}_{D}\;{+C}_{S}),$$
(6)$$C_{\mathrm{DS}}\;=\;C_{D}C_{S}{\;/\;(C}_{G}{+C}_{D}\;{+C}_{S}).$$

Figure 6 shows a typical impedance plot of a MOSFET small-signal equivalent circuit with predetermined parameters. A set of typical values for the parasitic inductances (*L*_G_ = 15 nH, *L*_D_ = 20 nH, and *L*_S_ = 30 nH), parasitic capacitances (*C*_S_ = 5 nF, *C*_D_ = 10 nF, and *C*_G_ = 15 nF), and resistances (*R*_G_ = 1.5 Ω, *R*_D_ = 0.5 Ω, and *R*_S_ = 0.5 Ω) is presented in the Advanced Design System (ADS) simulation setup with a frequency sweep from 1 to 300 MHz. The ADS simulated magnitude and phase angle of impedance (*Z*) parameters are shown in Figure 6a. *Z*_11_, *Z*_12_, *Z*_21_, and *Z*_22_ of the series *RLC* circuit can be calculated using Equations (7)–(9). The effect of capacitive reactance and resistance can be neglected at high frequency, *f*_High_ (endpoint of the frequency range). Therefore, the two-port network representation of the MOSFET equivalent circuit can be simplified as shown in Figure 6d. The high-frequency impedance is dominated by the inductive reactance, and the parasitic inductances *L*_S_, *L*_G_, and *L*_D_ can be calculated through Equations (10)–(12). At the *f*_SRF_, inductive and capacitive reactance cancel each other, and the impedance magnitude has its minimum value. The two-port network representation of the MOSFET equivalent circuit can be simplified as shown in Figure 6c. The parasitic resistances *R*_S_, *R*_G_, and *R*_D_ can be determined at the *f*_SRF_ through Equations (13)–(15). Meanwhile, the effect of inductive reactance and resistance can be neglected at low frequency, *f*_Low_ (starting point of the frequency range). Therefore, the two-port network representation of the MOSFET equivalent circuit can be simplified as shown in Figure 6b. The equivalent capacitances *C*_G_, *C*_S_, and *C*_D_ are respectively determined by plugging the extracted *L*_S_, *L*_G_, and *L*_D_ into Equations (16)–(18). Finally, the capacitor star connection is converted to a delta connection through Equations (4)–(6) to extract parasitic capacitances *C*_GS_, *C*_GD_, and *C*_DS_.
(7)$$Z_{11}{=X}_{L_{S}}{+X}_{L_{G}}{+X}_{R_{S}}{+X}_{R_{G}}{+X}_{C_{S}}{+X}_{C_{G}\;},$$
(8)$$Z_{12}{=Z}_{21}{=X}_{L_{G}\;}{+X}_{R_{G}}{+X}_{C_{G}}\;,$$
(9)$$Z_{22\;}{=X}_{L_{D}}{+X}_{L_{G}}{+X}_{R_{D}}{+X}_{R_{G}}{+X}_{C_{D}}{+X}_{C_{G}}\;,$$
(10)$$L_{S}{+L}_{G}{=imag\;(Z}_{11\_\mathrm{High}}{)\;/\;w}_{11\_\mathrm{High}}\;,$$
(11)$$L_{G}{=imag\;(Z}_{12\_\mathrm{High}}{)\;/\;w}_{12\_\mathrm{High}\;},$$
(12)$$L_{D}{+L}_{G}{=imag\;(Z}_{22\_\mathrm{High}}{)\;/\;w}_{22\_\mathrm{High}\;},$$
(13)$$R_{S}{+R}_{G}{=Z}_{11\_\min\;},$$
(14)$$R_{G}{=Z}_{12\_\min}\;,\;$$
(15)$$R_{D}{+R}_{G}{=Z}_{22\_\min\;},$$
(16)$$\frac{C_{S}C_{G}}{C_{S}{+C}_{G}}{=1\;/\;[w}_{11\_\mathrm{SRF}}^{2}\;\cdot {\;(L}_{S\;}{+L}_{G})]\;,$$
(17)$$C_{G}{=1\;/\;(w}_{12\_\mathrm{SRF}}^{2}\;\cdot {\;L}_{G})\;,$$
(18)$$\frac{C_{D}C_{G}}{C_{D}{+C}_{G}}{=1\;/\;[w}_{22\_\mathrm{SRF}}^{2}\;\cdot {\;(L}_{D\;}{+L}_{G})]\;.$$

## 3. Experimental Results and Discussion

### 3.1. Validation Parasitic Inductance Extraction Approach for MOSFET

A 1000 V Si MOSFET (IXFK32N100P in TO−247 package) and a 1200 V SiC MOSFET (C2M0160120D in TO−247 package) were used in this paper to verify the two-port network measurement technique. Figure 7 shows the schematic of the two-port network model and the VNA measurement system. An additional test fixture [30], which reserves three connection positions and ensures the low-inductance connections between VNA and terminals, must be designed to ensure the effective connection between MOSFET and VNA. The test fixture shall have a negligible loss, good impedance match (50 Ω), and high isolation between input and output. The printed circuit board (PCB) test fixture comprises two 50 Ω SMA adaptors, two 50 Ω microstrip lines, and a through-hole, as shown in Figure 7a. The top copper layer of the PCB is graphically processed into two 50 Ω microstrip lines. The bottom copper layer of the PCB is reserved for interconnection with the VNA ground. The MOSFET was installed on the PCB test fixture and connected with VNA through SMA. The MOSFET source terminal is interconnected with VNA port 1, the MOSFET drain terminal is interconnected with VNA port 2, and the gate terminal is interconnected with the VNA ground through a PCB through-hole. De-embedding calibration [31] was performed to remove the systematic errors caused by test cables, adapters, and fixtures before VNA measurement. The 80502D calibration kit (Keysight, Santa Rosa, CA, USA) provided by Keysight was used in this study to perform the short-open-load VNA calibration. A new “through” calibration element based on a PCB test fixture was designed to replace the “through” calibration element in the 80502D kit to extend the measurement plane from SAM coaxial connected to the interface of the device plane.

ADS circuit simulation was used to validate the proposed two-port parasitic inductance extraction methodology. First, the S−parameter of the MOSFET was obtained from VNA measurement and converted into Z−parameters. Then, the parasitic inductances, capacitances, and resistances were accurately calculated from *Z*−parameters through Equations (10)–(18). Finally, the extracted parasitic parameters were plugged back into the small-signal equivalent circuit of the power MOSFET for ADS simulation over a frequency range of 100 kHz to 400 MHz (the frequency range is not strictly fixed and can be adjusted according to different MOSFETs). Figure 8 shows the frequency response curves of the *Z*−parameters obtained from ADS simulation and VNA experimental measurement. The ADS simulation curve (red dashed line) was found to be in good agreement with the experimental value (black solid line) in Si and SiC MOSFETs, which indicates that the derived parasitic parameter extraction mathematical formulas (Equations (10)–(18)) and extraction methodology mentioned in Section 2.3 are effective and correct. A through-hole must be set on the PCB test fixture to connect the MOSFET gate terminal to the VNA ground. Unfortunately, through-hole will cause unwanted signal reflection on the transmission path, resulting in a slight impedance mismatch between the simulation and measurement of *Z*_12_ and *Z*_21_ at frequencies above the *f*_SRF_. *L*_S_ is calculated by subtracting Equation (10) from Equation (11). Thus, the gate parasitic inductance, *L*_G_, and the impedance mismatch introduced by through-hole can be excluded from the calculation. In addition, the secondary validation approach was realized by comparing the parasitic capacitances extracted from the proposed approach with the device datasheet values. The parasitic capacitances of the SiC MOSFET (C2M0160120D) obtained from the proposed two-port extraction technique were 0.49, 0.27, and 0.55 nF. Considering the unavoidable measurement error, the extracted capacitances were consistent with the datasheet values reported in [29] (*C*_GS_ = 0.47 nF, *C*_GD_ = 0.28 nF, and *C*_DS_ = 0.51 nF, *f* = 1 MHz), and the mismatch was 4.25%, 3.57%, and 7.84%. The experimental results show that the proposed two-port extraction methodology is suitable for accurately extracting the parasitic inductance of discrete-power MOSFETs.

### 3.2. Analysis of Parasitic Inductance with Bond Wire Fault

This paper aims to provide a precursor for bond wires’ degradation. The current study used the approach of cutting off bond wires to simulate their faults to shorten the duration of experimental tests. Laser equipment was used to remove the epoxy layer of the power device to expose the bond wires completely. The bond wire damage models were then established by manually cutting off the bond wires individually. The damage models of IXFK32N100P and C2M0160120D are shown in Figure 9.

When one bond wire is cut off (for instance, No. 1), the bond wire self-inductance *L*_1_ and mutual inductances *M*_1n_ and *M*_m1_ (*M*_12_, *M*_13_, *M*_14_, *M*_15_, and *M*_16_, and *M*_21_, *M*_31_, *M*_41_, *M*_51_, and *M*_61_, are generated by the magnetic coupling between bond wire No. 1 and other bond wires Nos. 2–6, respectively) no longer exist. The parasitic inductances of other valid bond wires (Nos. 2–6) are calculated by Equations (1)–(3), described in Section 2.2, and summarized in Table 3. The total parasitic inductance of IXFK32N100P is 4.51 nH, which is attributed to the result of the parallel connection of the five remaining bond wires. The total parasitic inductance of C2M0160120D is 4.26 nH, which is the result of the parallel connection of the three remaining bond wires. Similarly, for other bond wire damage models, such as cutting off two bond wires (Nos. 1 and 2), cutting off three bond wires (Nos. 1, 2, and 3), cutting off four bond wires (Nos. 1, 2, 3, and 4), and cutting off five bond wires (Nos. 1, 2, 3, 4, and 5), the parasitic inductances are calculated from Equations (1)–(3), and listed in Appendix A, Table A1, Table A2, Table A3 and Table A4. The corresponding relationship between parasitic inductance, *L*_Bond_, and the number of cutoff bond wires is shown in Figure 10. For ease of description, the liftoff of one or two bond wires is defined as “slight fault” and that of three or more bond wires is “serious fault.” Each of the total bond wire parasitic inductances of IXFK32N100P and C2M0160120D are positively correlated with the number of the bond wire cutoff. For slight fault, the percentage changes of *L*_Bond_IX_ in IXFK32N100P were 4.16% and 22.40%, while those of *L*_Bond_C2_ in C2M0160120D were 13.60% and 43.20%, showing high sensitivity. These results indicate that the proposed bond wire damage detection approach is reasonable. Figure 10 indicates that the parasitic inductance, *L*_Bond_, rises with the increase in the number of bond wire cutoffs, which can be potentially used as a precursor of bond wire damage. However, the parasitic inductance generated by the source terminal is not considered in the numerical analysis. The source parasitic inductance extracted from the two-port network measurement includes the bond wire parasitic inductance and the source terminal parasitic inductance. The sensitivity of MOSFET source parasitic inductance with various bond wire cutoffs is discussed in the following.

### 3.3. Bond Wire Experimental Results

The *S*−parameters of the bond wire damage models were measured with the VNA over a frequency range of 100 kHz to 400 MHz and then converted to *Z*−parameters. Figure 11 shows the *Z*−parameter frequency response curves of IXFK32N100P Si MOSFET with various bond wire faults. At frequencies above the *f*_SRF_, *Z*_11_ (*Z*_11_High_ = *L*_S_ + *L*_G_) increased with the number of bond wire cutoffs, whereas *Z*_12_ (*Z*_12_High_ = *L*_G_), Z_21_ (*Z*_21_High_ = *L*_G_), and Z_22_ (*Z*_22_High_ = *L*_G_ + *L*_D_) changed by less than 1 Ω when five bond wires were cutoff. This finding indicates a strong positive correlation between the damage of bond wires and the increase in source parasitic inductance, which is consistent with the presented theoretical expectation. The ratios expressed as percentage changes were calculated to define the degree of bond wire degradation. Percentage changes in *Z*_11_ and *L*_S_ are the different ratios of the measured value with the actual device compared with the initial value of the fault-free device under testing. For IXFK32N100P, the percentage changes in *Z*_11_ parameters of each bond wire damage model were 0.51%, 3.52%, 6.46%, 8.46%, and 28.95% at 400 MHz. For C2M0160120D, the Z−parameter frequency response curves are shown in Figure 12. This finding has a similar change trend as in Figure 11 with the increase in the number of bond wire cutoffs. The percentage changes of Z_11_ at 400 MHz were 2.46%, 5.60%, and 14.72%. 

Figure 13 and Figure 14 show that the source parasitic inductance increased with the increment of the number of the bond wire cutoffs, which is consistent with the change in *L*_Bond_ obtained by numerical calculation (Appendix A, Table A1, Table A2, Table A3 and Table A4). The parasitic inductance and resistance with various bond wire cutoffs are shown in Table 4. MOSFET is equivalent to the inductive element when the frequency is larger than the *f*_SRF_. With the increase in frequency, the skin effect forces the increase in parasitic resistance of the conductors (such as bond wire and terminal) and the reduction in parasitic inductance. For IXFK32N100P, the percentage change in the source parasitic inductance *L*_S_IX_ significantly increased with the increment of the number of bond wire cutoffs, and the difference reached the maximum at 400 MHz. With bond wire cutoffs varying from 1 to 5, the *L*_S_IX_ at 400 MHz increased from 8.52 to 8.61, 9.20, 9.71, 10.03, and 13.46 nH, and the percentage changes were 1.12%, 7.96%, 13.98%, 17.75%, and 58.02%, respectively. However, the extracted source parasitic resistance *R*_S_IX_ did not show the expected regular increase with the rising severity of bond wire faults. This phenomenon is due to the excessively low *f*_SRF_ of IXFK32N100P (≈9 MHz) to identify the *R*_S_IX_ (<0.015 Ω) effectively. Similarly, for C2M0160120D, the percentage change of source parasitic inductance *L*_S_C2_ has the maximum difference at 400 MHz. With bond wire cutoffs varying from 1 to 3, the *L*_S_C2_ at 400 MHz increased from 8.88 to 9.30, 9.85, and 11.43 nH, and the percentage changes were 4.71%, 10.97%, and 28.79%, respectively. SiC MOSFET has a higher switching frequency than traditional Si-based power semiconductor devices. The *f*_SRF_ of C2M0160120D is close to 58 MHz. Thus, the skin effect induced the increment of source parasitic resistance *R*_S_C2_ to more than 0.400 Ω, which reduced the identification accuracy requirements of *R*_S_C2_ by an order of magnitude, facilitating its accurate identification. The measurement results of the percentage change of *R*_S_C2_ with various bond wire cutoffs were 1.35%, 4.24%, and 17.21%, indicating that *R*_S_C2_ can be used as another precursor for bond wire fault detection. However, Table 4 shows that the resolution of *R*_S_C2_ was lower than that of *L*_S_C2_ in a two-port VNA measurement. If the *f*_SRF_ of MOSFET is low, then the influence of the skin effect on the conductor is not observed. Thus, the source parasitic resistance is too small to be accurately identified. Therefore, the source parasitic resistance can only be used to identify the bond wire damage of high switching frequency MOSFETs, especially in emerging wide-bandgap SiC MOSFETs.

The bond wire parasitic inductances obtained by numerical calculation and the source parasitic inductance extracted from VNA measurement are compared in Figure 15. The parasitic inductances obtained by the two approaches have a similar trend with the increase of the number of bond wire cutoffs. For the two MOSFETs, the difference in parasitic inductance between numerical calculation and VNA measurement was maintained at ∆*L*_C2_ = 4.1 ± 0.3 nH and ∆*L*_IX_ = 4.7 ± 0.3 nH, respectively. This difference can be attributed to the following two reasons: (1) VNA-measured Z−parameters include source terminal parasitic inductance, which is the main reason for the difference, as shown in Figure 15. (2) The spacing between two bond wires is taken as the average value to simplify the mutual inductance numerical calculation. Therefore, the VNA measurement extracted values are consistent with the numerically calculated values considering the fixed difference (∆*L*_C2_ and ∆*L*_IX_). The comparison results further verify the effectiveness of the proposed method of extracting the parasitic inductance from VNA measurement by considering a power MOSFET as a two-port network.

The change in source parasitic inductance can be accurately distinguished regardless of a “slight fault” or “serious fault.” Thus, the proposed bond wire damage detection approach in this paper has high discrimination. In addition, the parasitic parameters were extracted under zero DC biasing voltage (off-state) based on the two-port network VNA measurement, which can effectively avoid the design of additional test circuits, demonstrating its advantages compared with the traditional double-pulse power test. This approach offers an effective bond wire quality screening technology for power-discrete devices without power on treatment.

## 4. Conclusions

A novel bond wire damage detection approach on a MOSFET power device based on two-port network measurement by detecting parasitic inductance was proposed with theoretical analysis and experimental validation. The numerical calculation showed that the source parasitic inductance of discrete MOSFETs increased with the rising severity of bond wire faults, which can be used as a fault indicator to effectively determine bond wire liftoff faults. By considering a power device as a two-port network, MOSFET is equivalent to a pure inductance element at high frequency, and the parasitic inductances are accurately extracted from the Z−parameters without turning on MOSFETs. The experimental results indicated that the source parasitic inductance, *L*_S_, increased with the fault number of bond wires, and even the slight fault showed high sensitivity, which can effectively quantify the number of bond wire liftoffs of discrete MOSFETs under zero biasing conditions. Meanwhile, the feasibility and applicability of using source parasitic resistance, *R*_S_, to identify the bond wire fault were discussed. The skin effect forced the increase in bond wire parasitic resistance to an effective detection scale due to the higher *f*_SRF_, which was significantly positively correlated with the severity of bond wire faults. However, as a failure precursor, parasitic resistance is suitable for the detection of high switching frequency MOSFETs, especially in emerging wide-bandgap SiC MOSFETs, and the recognition resolution of low-frequency power MOSFETs was insufficient. The proposed two-port network VNA measurement approach was impressively achieved without turning on the MOSFET. Thus, designing additional test circuits and controlling the junction temperature is unnecessary. This approach offers an effective bond wire fault detection technology for power devices and can be extended by establishing online quality monitoring technology in future research.

## Figures and Tables

**Figure 1 micromachines-13-01075-f001:**
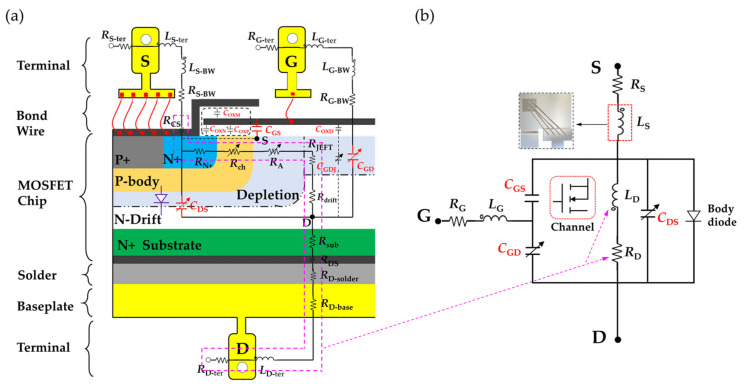
(**a**) Cross-section of a power MOSFET, and (**b**) small-signal equivalent circuit.

**Figure 2 micromachines-13-01075-f002:**
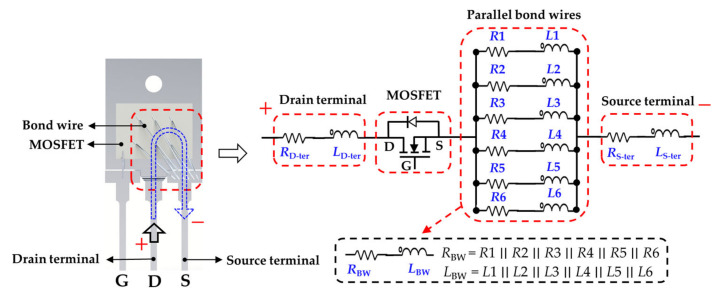
Simplified *RL* equivalent of a MOSFET in TO−247 package.

**Figure 3 micromachines-13-01075-f003:**
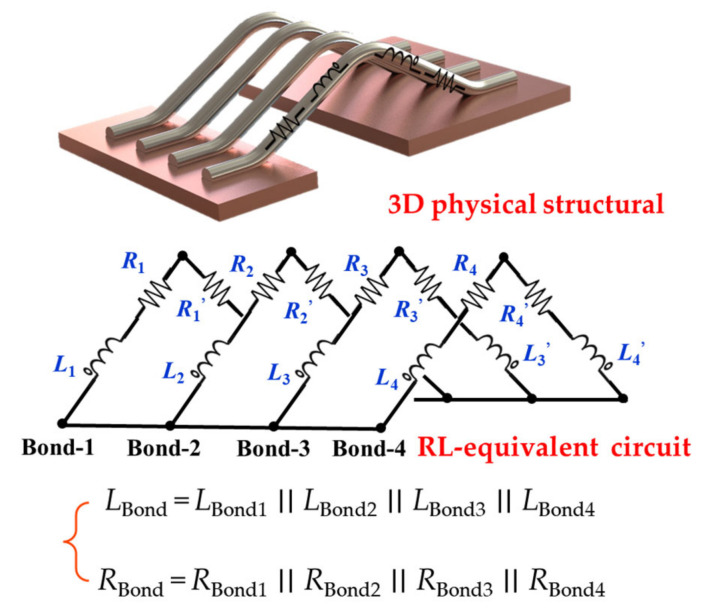
3D structural diagram and *RL*-equivalent circuit of parallel bond wires.

**Figure 4 micromachines-13-01075-f004:**
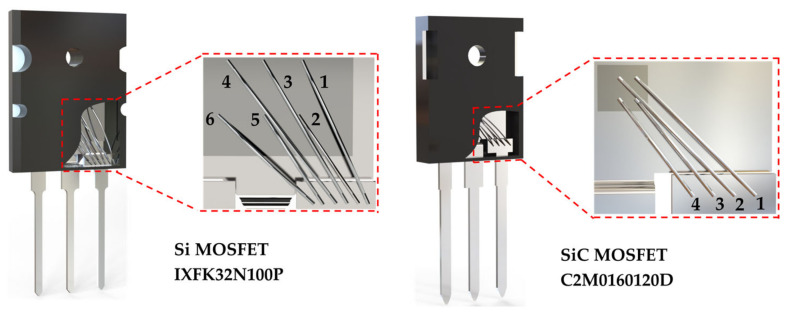
Internal structure of IXFK32N100P and C2M0160120D in TO−247, and the serial number marked on each bond wire.

**Figure 5 micromachines-13-01075-f005:**
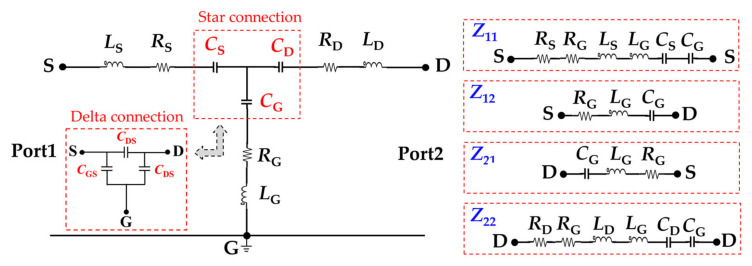
Two-port network of a MOSFET small-signal circuit model under zero biasing condition.

**Figure 6 micromachines-13-01075-f006:**
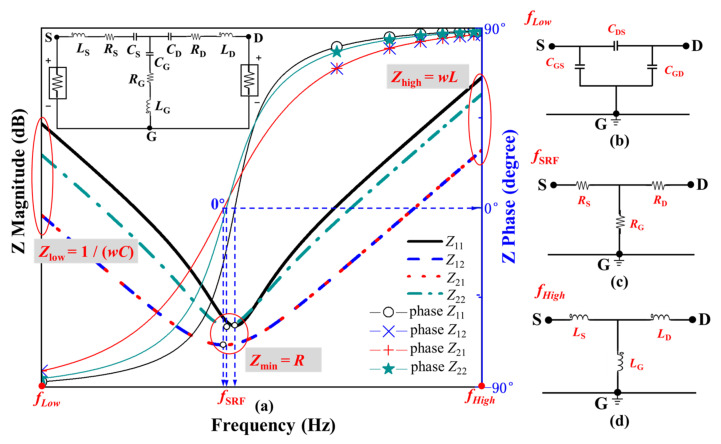
MOSFET parasitic parameter extraction theory. (**a**) Impedance magnitude and phase curves of a typical MOSFET. (**b**) Two-port network representation for the MOSFET at low frequency. (**c**) Two-port network representation for the MOSFET at the *f*_SRF_. (**d**) Two-port network representation for the MOSFET at high frequency.

**Figure 7 micromachines-13-01075-f007:**
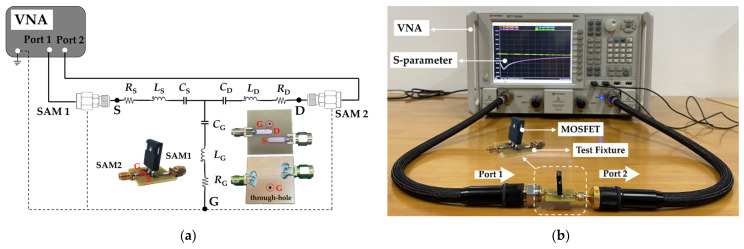
Two-port VNA measurement setup for MOSFET parasitic inductance extraction. (**a**) Schematic of proposed two-port network extraction approach. (**b**) VNA measurement setup including the PCB fixture.

**Figure 8 micromachines-13-01075-f008:**
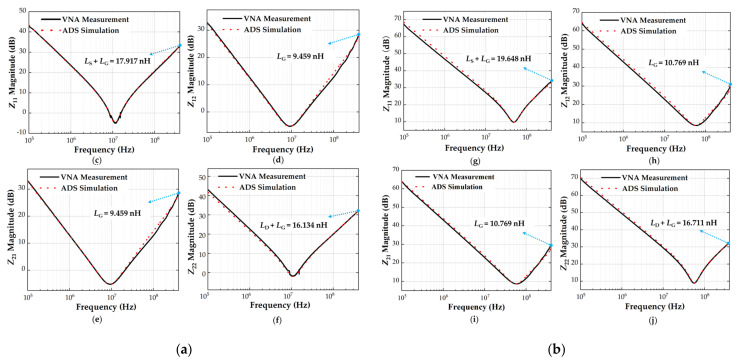
Z−parameters obtained from experimental measurement and ADS simulation. (**a**) Si MOSFET IXFK32N100P in TO−247 package ((**c**) Z_11_, (**d**) Z_12_, (**e**) Z_21_, (**f**) Z_22_). (**b**) SiC MOSFET C2M0160120D in TO−247 package ((**g**) Z_11_, (**h**) Z_12_, (**i**) Z_21_, (**j**) Z_22_).

**Figure 9 micromachines-13-01075-f009:**
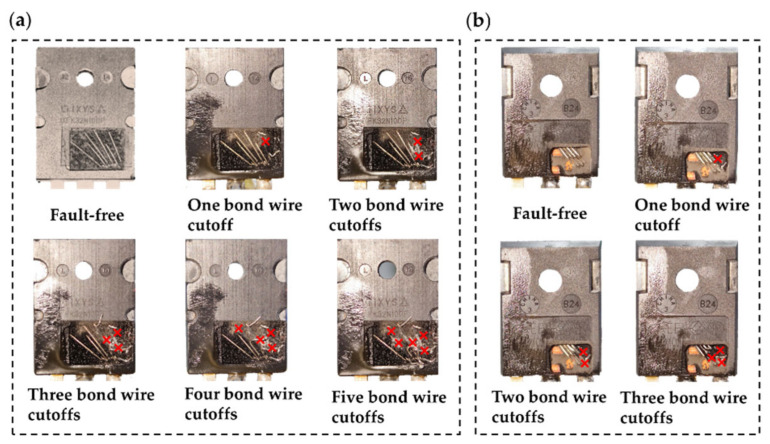
Bond wire damage models. (**a**) TO−247 Si MOSFET (its source has six aluminum bond wires connected in parallel). (**b**) TO−247 SiC MOSFET (its source has four aluminum bond wires connected in parallel).

**Figure 10 micromachines-13-01075-f010:**
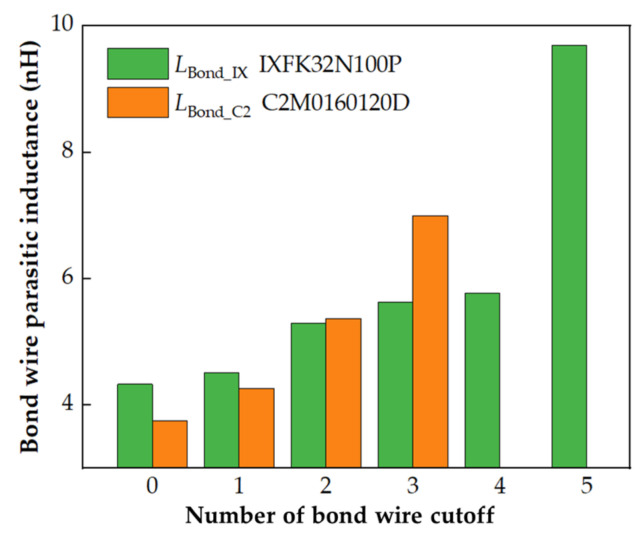
Parasitic inductance with different cutoff numbers of bond wires.

**Figure 11 micromachines-13-01075-f011:**
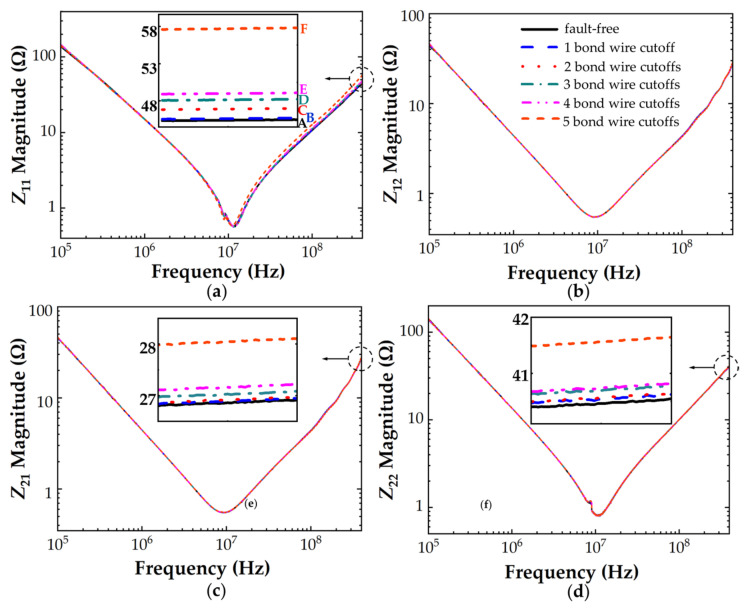
*Z*−parameters of IXFK32N100P with different cutoff numbers of bond wires (A = 45.03 Ω, B = 45.30 Ω, C = 46.62 Ω, D = 47.94 Ω, E = 48.84 Ω, F = 58.07 Ω). (**a**) Z_11_, (**b**) Z_12_, (**c**) Z_21_, (**d**) Z_22_.

**Figure 12 micromachines-13-01075-f012:**
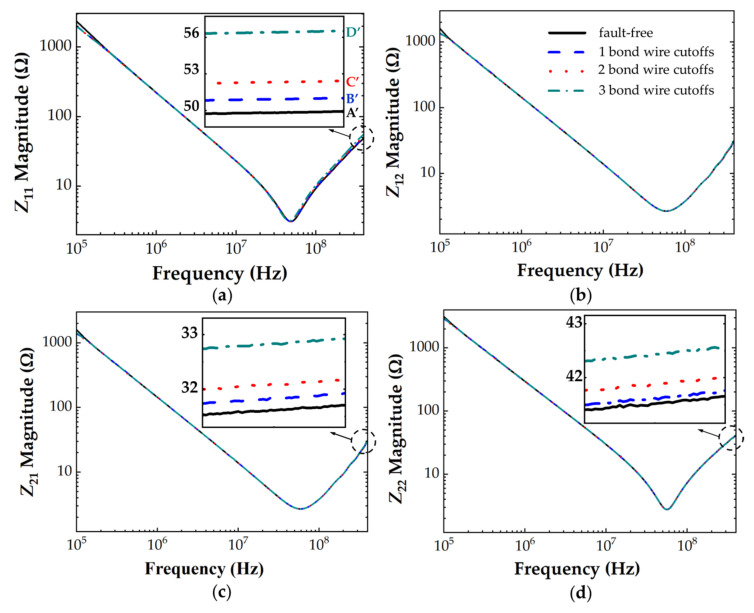
*Z*−parameters of C2M0160120D with different cutoff numbers of bond wires (A’ = 49.38 Ω, B’ = 50.60 Ω, C’ = 52.15 Ω, D’ = 56.66 Ω). (**a**) Z_11_, (**b**) Z_12_, (**c**) Z_21_, (**d**) Z_22_.

**Figure 13 micromachines-13-01075-f013:**
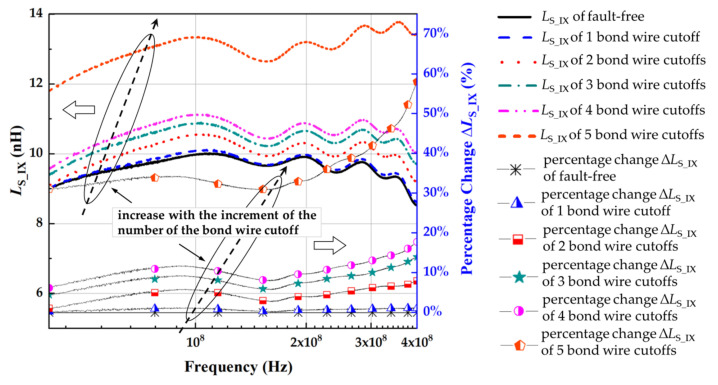
*L*_S_IX_ of IXFK-32N100P with different cutoff numbers of bond wires.

**Figure 14 micromachines-13-01075-f014:**
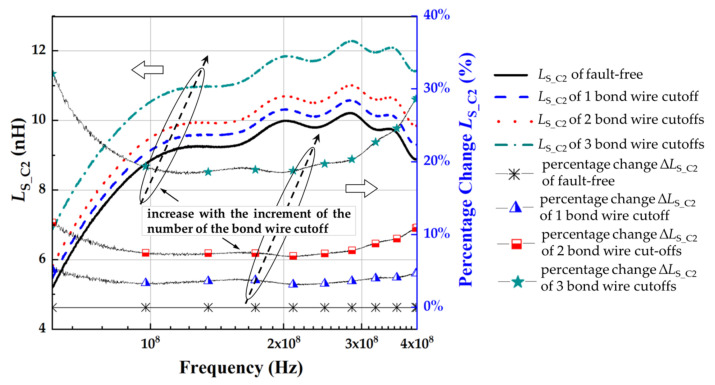
*L*_S_C2_ of C2M0160120D with different cutoff numbers of bond wires.

**Figure 15 micromachines-13-01075-f015:**
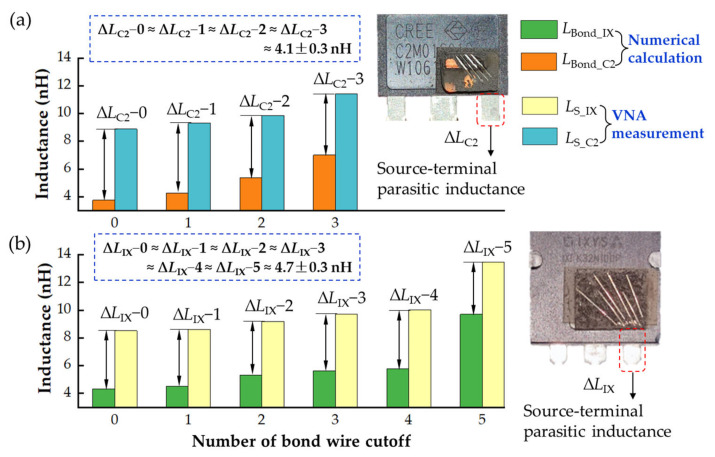
Numerical calculation and VNA measurement of parasitic inductance with different cutoff numbers of bond wires. (**a**) Comparison results of C2M0160120D. (**b**) Comparison results of IXFK32N100P.

**Table 1 micromachines-13-01075-t001:** Bond wire dimension for MOSFETs.

Bond Wire No.		1	2	3	4	5	6
IXFK32N100P	*l* (mm)	13.54	8.76	14.16	15.50	8.94	11.35
	*d* (mm)	0.279					
C2M0160120D	*l* (mm)	5.78	5.90	7.85	8.00		
	*d* (mm)	0.178					

**Table 2 micromachines-13-01075-t002:** Parasitic inductance for IXFK32N100P and C2M0160120D discrete MOSFETs.

Bond Wire No.	1		2	3	4	5	6	*L*_Bond_No_. nH
Si MOSFET IXFK32N100P	1	12.04	3.4	4.66	3.76	2.16	2.08	28.1
	2	3.40	7.04	3.66	3.40	2.39	1.86	21.75
	3	4.66	3.66	12.72	4.98	3.51	2.58	32.11
	4	3.76	3.4	4.98	14.19	3.76	3.57	33.66
	5	2.16	2.39	3.51	3.76	7.22	3.22	22.26
	6	2.08	1.86	2.58	3.57	3.22	9.7	23.01
	*L* _Bond_IX_							4.33
SiC MOSFET C2M0160120D	1	5.63	3.11	3.11	1.85			13.70
	2	3.11	5.83	3.24	2.40			14.59
	3	3.11	3.24	6.78	3.84			16.99
	4	1.85	2.40	3.84	7.00			15.08
	*L* _Bond_C2_							3.75

**Table 3 micromachines-13-01075-t003:** Parasitic inductance, *L*_Bond_, with one bond wire cutoff.

Bond Wire No.	1		2	3	4	5	6	*L*_Bond_No_. nH
Si MOSFET IXFK32N100P	1	/						
	2	/	7.04	3.66	3.4	2.39	1.86	18.35
	3		3.66	12.72	4.98	3.51	2.58	27.45
	4		3.4	4.98	14.19	3.76	3.57	29.9
	5		2.39	3.51	3.76	7.22	3.22	20.1
	6		1.86	2.58	3.57	3.22	9.7	20.93
	*L* _Bond_IX_							4.51
SiC MOSFET C2M0160120D	1	/						
	2	/	5.83	3.24	2.40			11.47
	3		3.24	6.78	3.84			13.87
	4		2.40	3.84	7.00			13.24
	*L* _Bond_C2_							4.26

**Table 4 micromachines-13-01075-t004:** Parasitic inductance and resistance with different cutoff numbers of bond wires.

Bond Wire Cutoff Model	Si MOSFET				SiC MOSFET			
	400 MHz		*f* _SRF_		400 MHz		*f* _SRF_	
	*L*_S_ nH	Percentage Change %	*R*_S_ Ω	Percentage Change %	*L*_S_ nH	Percentage Change %	*R*_S_ Ω	Percentage Change %
0	8.52	0	0.012	0	8.88	0	0.401	0
1	8.62	1.12	0.014	16.67%	9.30	4.71	0.406	1.25%
2	9.20	7.96	0.013	8.33%	9.85	10.97	0.418	4.24%
3	9.71	13.98	0.014	−8.33%	11.43	28.79	0.470	17.21%
4	10.03	17.75	0.006	−50.00%				
5	13.46	58.02	0.029	141.67%				

## Data Availability

Not applicable.

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
