# Peer review of "Bond Wire Damage Detection Method on Discrete MOSFETs Based on Two-Port Network Measurement"

_micromachines, 2022, doi:10.3390/mi13071075_

Round 1

Reviewer 1 Report

The authors used a two-port network measurement to detect bond wire damage by measuring the MOSFET source parasitic inductance (LS). It might be an approach to judge the quality of bonding Power Device. However, the cost for a two-port network measurement might not be acceptable and not for mass production under a real test. Here are some comments for the authors.

1.Please describe Table I and II more precisely.

2.Reference 5 is a very important one, that mentioned the package failure rate. However, it was published in 1996. Nowadays, the package technology is mature enough. I don't think the failure rate of package will be large than 1%. The author should use a recently published literature.

3.For a two-port network measurement system, the cost might be a big issue. In addition, the throughput for testing is also a big problem.  

4.For a two port network measurement, how do you make sure your ground locate at the source?  Please take a top view photograph of your test board. By the way, how do you deduct the the signal lost at drain side and TO247 package?

5.Please modify the area of fig. 12.

Reviewer 2 Report

The paper proposed a detection method with two-port network measurement to obtain Z parameters to further extract the inductance to identify bond wire damage. With this method, two types of power Mosfet devices are tested and analyzed with cutting off of bond wire. However, there are some issues in the text that need to be addressed before considering publication.

1Are the curves in Fig 8 labeled with ADS simulation calculated according to the equivalent circuit model in Fig 5? If yes, how are the parameters such as CsLsRs  obtained in the circuit model? If these parameters are appropriated with the VNA test results, whats the intention of theses curves in Fig8? (line 238)

2Why does the frequency range from 100KHz to 0.4GHz is selected? How are high and low frequencies defined(line 191)

3) The description of the Fig 10(curve) and the Fig 15(bar chart) are not clear , especially the comparison between them, its confusing. Please modify.

4Please confirm the expression of formula 1 and formula 2 , the diameter “d“” in the text is not mentioned in the formula(line 133). It would be better to offer the specific information about diameter and length of the wire bond.

5) To better showcase the merit of the proposed method with two port VNA measurement to extract inductance against the method mentioned in the introduction (voltage precursor-based approach and current precursor-based approach), it is recommended to give quantitative analysis.

6) There is a mistake in caption of Figure 6a. The phase is not shown in the figure.line 192

Round 2

Reviewer 1 Report

This version is fine. It can be accepted for publication.